# Current Knowledge and Prospects for Renal Hemangioblastoma and Renal Cell Carcinoma with Hemangioblastoma-like Features

**DOI:** 10.3390/biomedicines11051467

**Published:** 2023-05-17

**Authors:** Fumiyoshi Kojima, Fidele Y. Musangile, Ibu Matsuzaki, Kenji Yorita, Naoto Kuroda, Yoji Nagashima, Shin-ichi Murata

**Affiliations:** 1Department of Human Pathology, Wakayama Medical University, 811-1 Kimiidera, Wakayama 641-8509, Japan; fidelemusangile@gmail.com (F.Y.M.); m_ibu@wakayama-med.ac.jp.jp (I.M.); smurata@wakayama-med.ac.jp (S.-i.M.); 2Department of Diagnostic Pathology, Japanese Red Cross Kochi Hospital, 1-4-63-11 Hataminami-cho, Kochi-shi, Kochi 780-8562, Japan; kenjiyorita@gmail.com; 3Department of Internal Medicine, Kinrou Hospital, 3-2-28 Azounokitamachi, Kochi 781-0011, Japan; kurochan@yahoo.co.jp; 4Department of Surgical Pathology, Tokyo Women’s Medical University Hospital, 8-1 Kawada-cho, Shinjuku-ku, Tokyo 162-8666, Japan; nagashima.yoji@twmu.ac.jp

**Keywords:** hemangioblastoma, hemangioblastoma-like, renal cell carcinoma, clear cell, hemangioma, fibromyomatous stroma, leiomyomatous stroma, VHL, TSC, MTOR

## Abstract

Tumors exhibiting histopathological findings similar to those of hemangioblastoma of the central nervous system (CNS-HB) rarely develop in the kidneys. Currently, renal hemangioblastoma (RHB) is considered analogous to CNS-HB; however, they differ in gross appearance, as well as immunohistochemical and molecular findings. In contrast, some renal cell carcinomas reportedly comprise distinct, clear cell renal cell carcinoma (CCRCC)- and hemangioblastoma (HB)-like areas. Initially, renal cell carcinomas with HB-like features (RCC-HBs) were considered a morphological variant of CCRCC owing to their diverse histological findings. However, the immunohistochemical and molecular findings of RCC-HBs suggest that RCC-HB is distinct from CCRCC. Additionally, one of the RCC-HBs had a focal leiomyomatous stroma and *TSC2* variant, suggesting that RCC-HB and RCC with fibromyomatous stroma (RCC-FMS) might belong to the same disease entity. Therefore, we comprehensively reviewed the clinical, pathological, and molecular features of RHB, RCC-HB, and the related tumors and discussed the similarities, differences, and relationships between them. We believe that our review would serve as a foundation for further investigation on elucidating the relationship between CNS-HB, RHB, RCC-HB, and RCC-FMS.

## 1. Introduction

Hemangioblastoma (HB) is a benign tumor that develops in the central nervous system (CNS), particularly in the cerebellum. Approximately 25% of CNS-HB cases are associated with Von Hippel–Lindau disease (VHL) [1]. *VHL* alterations are also strongly implicated in the pathogenesis of sporadic HBs [2]. CNS-HB occurs equally in men and women. CNS-HB often presents as a well-circumscribed cystic mass composed of stromal cells and abundant vascular proliferation. CNS-HB is particularly characterized by stromal cells which have lipid-containing cytoplasmic vacuoles and exhibit immunopositivity for inhibin-α and S100 [2].

Tumors exhibiting the histological and immunohistochemical features of CNS-HB rarely develop in extra-neuraxial sites, including the soft tissue, retroperitoneum, bone, skin, liver, pancreas, intestine, and kidneys [1,3,4,5]. Among extra-neuraxial sites, renal HBs (RHBs) have been frequently reported with studies focusing on the potential diagnostic pitfall of misdiagnosing RHB as clear cell renal cell carcinoma (CCRCC) because of the histological similarity between them [3,6]. Currently, RHB is considered to be the same disease as CNS-HB; however, RHBs have different macroscopic, immunohistochemical, and molecular findings from CNS-HBs [7,8]. Recently, a few cases of renal cell carcinoma with HB-like features (RCC-HB) composed of CCRCC- and HB-like components have been reported [9,10]. The relationship between RHB and RCC-HB has not been clarified because RCC-HB occurs rarely. Initially, RCC-HB was regarded as a morphological variant of CCRCC, because CCRCC has diverse histological findings, including hemangioma-like features [11]. However, the hypothesis becomes less persuasive, as data on RCC-HBs have accumulated. Recently, a case of RCC-HB was reported to have fibromyomatous stroma (FMS) and a *TSC2* mutation without *VHL* mutation [12]. It suggested that there was some relationship between RCC-HB and renal cell carcinoma with FMS (RCC-FMS). Hence, in this review, based on the knowledge accumulated in recent years, we discuss the clinical, pathological, and molecular features of RHB, RCC-HB, and RCC-FMS and focused on the similarities, differences, and the relationship between them.

## 2. Renal Hemangioblastoma

RHB was first reported by Nonaka et al. in 2007 [1]. Approximately 30 RHBs have since been reported in the literature written in English [1,3,4,6,7,8,13,14,15,16,17,18,19,20,21,22,23]. RHBs exhibit similar histological findings to CCRCCs which exhibit solid growth of clear to pale tumor cells with abundant cytoplasm, thereby leading to RHB underdiagnosis or misdiagnosis as CCRCCs [3,6]. RHBs are detected equally in men and women. Most RHBs are detected incidentally during regular medical check-ups, whereas some patients with RHBs have complained of hematuria. The size of RHBs was relatively small, and all patients with RHB were alive after resection without recurrence or metastasis.

Macroscopically, RHB is a well-circumscribed solid tumor, with or without a capsule (Figure 1a). RHBs have hypercellular and hypocellular regions. The hypercellular regions comprise clear to eosinophilic, round-to-polygonal stromal cells with sheet-like or solid growth and a rich vasculature (Figure 1b). Most stromal cells had round-to-oval nuclei, but some stromal cells had pleomorphic or bizarre nuclei, and intranuclear cytoplasmic inclusions (Figure 1c,d). Most RHBs had hyaline globules. Stromal cells had vacuoles, suggesting cytoplasmic lipid accumulation (Figure 1e). The paucicellular area was fibrous, sparsely vascularized, with scattered stromal cells, and hyalinized (Figure 1f).

Immunohistochemical analysis revealed that most RHBs were positive for inhibin-α, S100, neuron-specific enolase, vimentin, and CAIX (Figure 2a,b). Approximately 50% of RHBs were focally positive for CD10 and PAX8 (Figure 2c,d), whereas some RHBs were focally positive for pan-cytokeratin (CK) and epithelial membrane antigen (EMA). However, all the RHBs were negative for the renal cell carcinoma (RCC)-marker and CK7.

Significant genetic alterations have not been reported in RHB, although a few molecular analyses, mainly targeting *VHL*, were performed. Furthermore, RHBs did not exhibit 3p loss of heterozygosity (LOH) or hypermethylation of the *VHL* promoter region in some cases. Therefore, we speculated that *VHL* alterations were not involved in RHB development.

It is important to differentiate RHB from CCRCC because of the differences in patient management. The differentiation between stromal cell predominant RHB and CCRCC is difficult. Differentiation will require the use of all available findings. Macroscopically, RHBs do not exhibit a golden-yellow color. Microscopically, there are stromal cells that are pale, foamy, and have multi-vacuolated cytoplasm with a bizarre nucleus, cytoplasmic intranuclear inclusion, or hyaline globules. Those worrisome histological findings also suggest RHB and immunostaining for inhibin-α and S100 are useful for differentiation.

## 3. Renal Cell Carcinoma with Hemangioblastoma-like Features

In 2015, Montironi et al. reported two RCC-HBs composed of distinct CCRCC- and HB-like areas [9]. To date, three additional RCCs, supposedly the same type of lesion as those in Montironi, have been reported in the English-language scientific literature: CCRCC-HB, RCC with hemangioma-like features, and RCC-HB with leiomyomatous stroma [10,12,24].

Among the five patients with RCC-HB, two were men and three were women. None of the patients with RCC-HB presented with signs of VHL or had a significant family history of VHL or tuberous sclerosis complex. Patients with RCC-HB were asymptomatic, had hematuria, or experienced discomfort in the kidney. The tumor size was relatively small (1.7–5.5 cm). All RCC-HBs, excluding one with focal cystic changes, were solid masses. All RCC-HBs were pT1 except for one RCC-HB with pT3a staged disease, and recurrence or death from the disease has not yet been reported since the follow-up period was insufficient for assessing long-term outcomes.

RCC-HBs were well-circumscribed tumors (Figure 3a). RCC-HBs had two distinct areas of CCRCC- and HB-like areas. Most RCC-HBs gradually transitioned from a CCRCC-like area to an HB-like area (Figure 3b). The HB-like area occupied a relatively large area of the tumor (30–70%). Additionally, most RCC-HBs revealed foci of regressive paucicellular areas. In the CCRCC-like area, the tumors exhibited sheet-like, solid, acinar, tubular, or papillary growth (Figure 3c). Tumors were composed of clear to eosinophilic cells with a low nuclear grade (1–2 (WHO/ISUP)) (Figure 3d). One RCC-HB patient had focal nucleoli equivalent to grade 3 (WHO/ISUP). In the HB-like area, vascular proliferation was conspicuous, stromal-like clear cells were intimately intermixed with thin-walled vessels, and the tumor cells were indistinct (Figure 3e,f). Some tumor cells had hyaline globules or vacuoles, whereas a few, scattered tumor cells had pleomorphic or bizarre nuclei (Figure 3g). One RCC-HB patient had a focal leiomyomatous stroma, mainly at the periphery of the tumor.

Immunohistochemically, the tumor cells were positive for PAX8, CAIX, CD10, EMA, pan-CK, RCC marker, and CK7 and negative for AMACR, inhibin-α, and S100 in the CCRCC-like area (Figure 4a,b). Tumor cells were positive for inhibin-α and S100 and negative for CK7 in the HB-like area (Figure 4c,d). For other antibody staining, tumor cells in the HB-like area revealed an immunoprofile similar to that of the CCRCC-like area.

Although molecular analyses had not been performed for RCC-HB, Kong et al. presented the first report on the whole exon sequencing of a RCC-HB in 2021 [12]. The authors detected a likely pathogenic missense single-nucleotide variant in *TSC2* (c.T311C, p.L104P) and an unknown significant missense single-nucleotide variant in *SETD2* (c.C721G, p.P241A). According to the molecular findings, RCC-HB is not a morphological variant of CCRCC and is distinct from CCRCC because of the lack of *VHL* mutations commonly harbored in CCRCC.

## 4. Renal Cell Carcinoma with Fibromyomatous Stroma

RCCs of clear cell type characterized with FMS had been reported under various names: RCC with smooth muscle stroma, renal adenomyomatous tumor (RAT), RCC with angioleiomyoma-like stroma, and RCC with leiomyomatous stroma [25,26,27,28]. Some RCCs, including CCRCCs and clear cell papillary renal cell tumors (CCPRCTs), have FMS. Currently, most RATs are considered to be within the same spectrum of CCPRCT. However, RCC-FMSs, excluding CCRCC and CCPRCT using immunohistochemical and/or molecular analysis, have unique and common histological, immunohistochemical, and molecular findings [25]. Based on the unique features, the WHO 2016 classification named them RCC with (angio)leiomyomatous stroma and classified them as an emerging/provisional entity. In 2021, the Genitourinary Pathology Society (GUPS) renamed RCC with (angio)leiomyomatous stroma as RCC-FMS and classified them as a novel entity [29]. For diagnosing RCC-FMS, GUPS emphasized diffuse CK7 positivity and required excluding CCRCC and CCPRCT. RCC-FMSs include various molecular abnormalities: *TSC1*, *TSC2*, *MTOR*, and *ELOC* (*TCEB1*) mutations frequently associated with monosomy 8. Some RCC-FMSs were associated with tuberous sclerosis complex. In the latest WHO 2022 classification, only *ELOC*-mutated RCC was separated from RCC-FMS and classified as a distinct subtype. RCC-FMSs with MTOR/TSC pathway alterations were not accepted as distinct entities [30].

RCC-FMS is a well-circumscribed tumor with a thick fibromuscular capsule. The tumor is traversed by the fibromuscular bundle extending from the capsule and exhibits its characteristic multiple nodular appearance (Figure 5a). The proportion of FMS in the tumor depends on the case, ranging from scant to abundant. RCC-FMSs exhibited elongated and branching tubular growth with foci of papillary formation (Figure 5b). The tumor cells are clear to small with low-grade nuclei. Some RCC-FMSs have foci of abundant vascular proliferation [26]. RCC-FMSs were immunohistochemically positive for CK7, CD10, and CK34βE12 and negative for AMACR, and exhibited focal cup-shaped CAIX positivity (Figure 5c,d). RCC-FMS is not reported to have HB-like features, including immunopositivity for inhibin-α. RCC-FMSs did not harbor 3p LOH or *VHL* mutations, unlike CCRCC. RCC-FMS often exhibited MTOR/TSC pathway alterations.

## 5. Relationship between RHB, RCC-HB, and Related Tumors

### 5.1. Similarities and Differences between RHB and CNS-HB

RHB and CNS-HB develop equally in men and women. CNS-HBs that are especially stromal cell predominant and RHB have similar histopathological findings. However, CNS-HB often exhibits a cystic mass, whereas RHBs are solid masses without cyst formation. Furthermore, they differ in their immunohistochemical features: typically, CNS-HB is negative for pan-CK, EMA, PAX8, and CD10, whereas RHB is often positive. Additionally, *VHL* alterations are predominantly involved in the pathogenesis of CNS-HB. However, molecular analyses for RHBs have suggested that *VHL* alterations are not involved in its pathogenesis. Therefore, whether RHB is an analogous tumor to CNS-HB might be debatable.

### 5.2. Similarities and Differences between RHB and RCC-HB

Both RHBs and RCC-HBs were detected equally in men and women. Patients with RHB or RCC-HB do not exhibit any signs of VHL. Both the RHBs and RCC-HBs were solid masses without cysts. The most distinct histological finding was that RCC-HB had distinct CCRCC-like components with apparent epithelial features: acinar or glandular architecture showing rigorous cell-to-cell connection. Immunohistochemical positivity for epithelial markers was weaker in RHB than in RCC-HB. However, most RHBs were positive for PAX8, CD10, EMA, and CAIX, similar to RCC-HBs. We speculate that *VHL* alterations are not involved in the pathogenesis of both tumors. We lack sufficient evidence to conclude whether or not RCC-HB is the same disease as RHB. The proportion of CCRCC-like components is often small, so the component can be easily missed or regarded as a collision between RHB and CCRCC. Some studies previously reported that RHBs are possibly RCC-HB and may have been underdiagnosed [19]. The differentiation between RHB and RCC-HB is important for patient management because the former is indolent, but the latter has malignant potential since one reported RCC-HB involved the renal vein.

### 5.3. Similarities and Differences between RCC-HB and RCC-FMS

It has not been reported that RCC-FMS have HB-like features. However, some reports described RCC-FMSs with hemangioma-like foci [26]. Additionally, Kong et al. reported a case of RCC-HB with FMS that harbored a TSC2 variant. According to the GUPS guidelines, the tumor can be classified as RCC-FMS. We cannot conclude whether RCC-HB is a member of RCC-FMS or whether RCC-HB is a tumor distinct from RCC-FMS in one case. However, considering that some RCC-FMSs have hemangioma-like foci, we speculate that RCC-HB is a tumor on the same spectrum as RCC-FMS, and RCC-HB might be a molecular variant of RCC-FMS with HB-like features and unremarkable FMS. However, the fact that FMS was mentioned in only one RCC-HB case contradicts the hypothesis that RCC-HB and RCC-FMS are included in the same disease entity. Additionally, if RCC-HBs without FMS are included in RCC-FMS, then FMS is a characteristic but not a requisite feature for diagnosing RCC-FMS. RCC-HB might prompt the modification of the disease concept of RCC-RMS and might expand the histological picture of RCC-FMS. Further research is required to establish the disease concept of RCC-FMS. It is almost impossible to differentiate RCC-FMS from CCRCC and CCPRCT in routine practice using hematoxylin and eosin-stained slides and immunohistochemistry, without molecular analyses. The HB-like features can be a clue to differentiate RCC-FMS from CCRCC.

## 6. Conclusions

In conclusion, we regarded RHB as an analogous tumor to CNS-HB, and RCC-HB as a morphological variant of CCRCC. However, increasing reports on RHB and RCC-HB have suggested that these tumors have unique histological, immunohistochemical, and molecular features (Table 1). Renal neoplasms with HB-like features, including RHB and RCC-HB, are currently gaining attention, especially the relationship between RCC-HB and RCC-FMS. To clarify the relationship between RHB, RC-HB, RCC-FMS, and CNS-HB, we needed to accumulate further RHBs and RCC-HBs and perform molecular analyses including *VHL* and the MTOR/TSC pathway for a large number of cases, in comparison with RCC-FMS and CNS-HB.

## Figures and Tables

**Figure 1 biomedicines-11-01467-f001:**
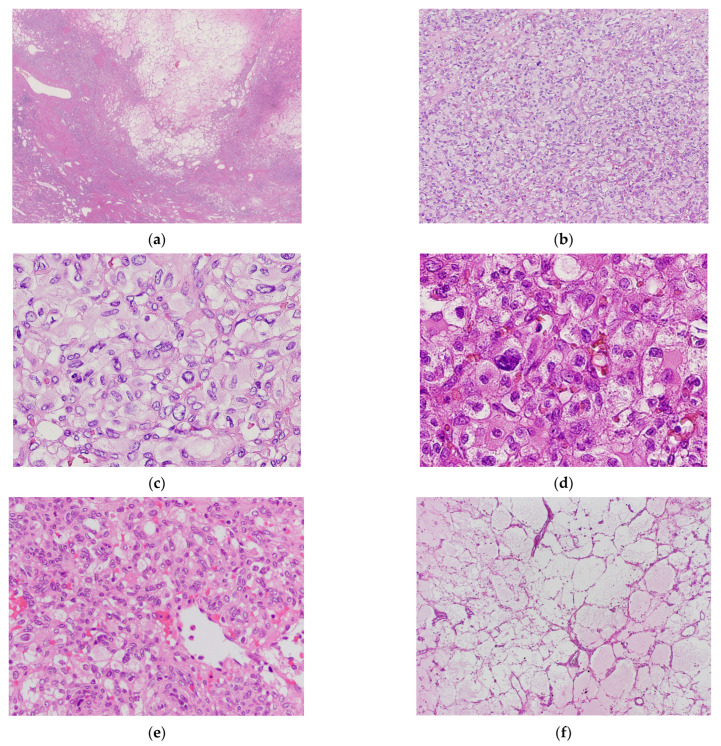
Histological findings of renal hemangioblastoma. (**a**) Whole slide imaging suggests that the tumor is well-circumscribed without a capsule. The tumor comprises cellular and edematous paucicellular areas. (**b**) At low-power magnification, tumor cells exhibit sheet-like growth with rich vasculature. (**c**) At high-power magnification, tumor cells exhibit pale, abundant cytoplasm with size variation. The nuclei are small and large. (**d**) At high-power magnification, a bizarre nucleus is scattered. (**e**) Some tumor cells have multivacuoles (high-power magnifiaction). (**f**) Paucicellular area is edematous, and vessels are scattered without stromal cells (low-power magnification).

**Figure 2 biomedicines-11-01467-f002:**
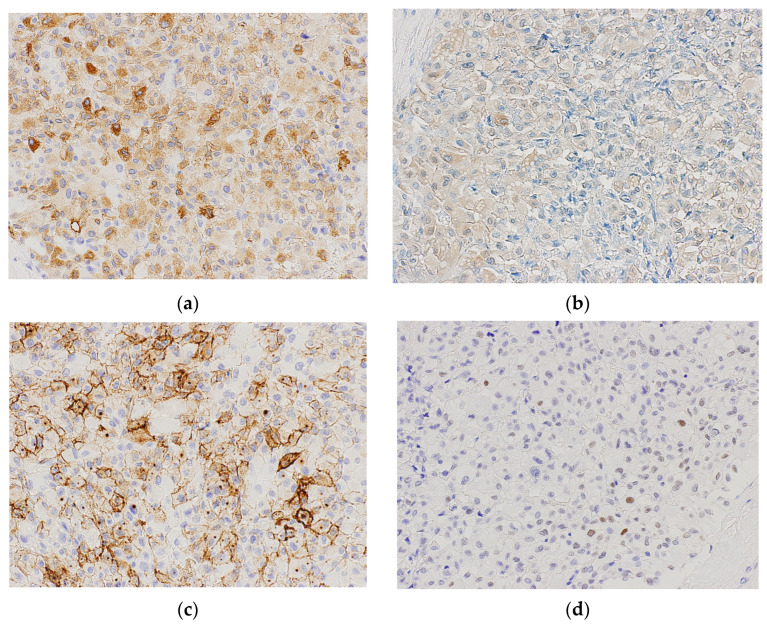
Immunohistochemical findings of renal hemangioblastoma. Tumor cells are positive for inhibin-α (**a**) and S100 (**b**) and focally positive for CD10 (**c**) (200× magnification). PAX8-positive tumor cells are scattered (**d**) (200× magnification).

**Figure 3 biomedicines-11-01467-f003:**
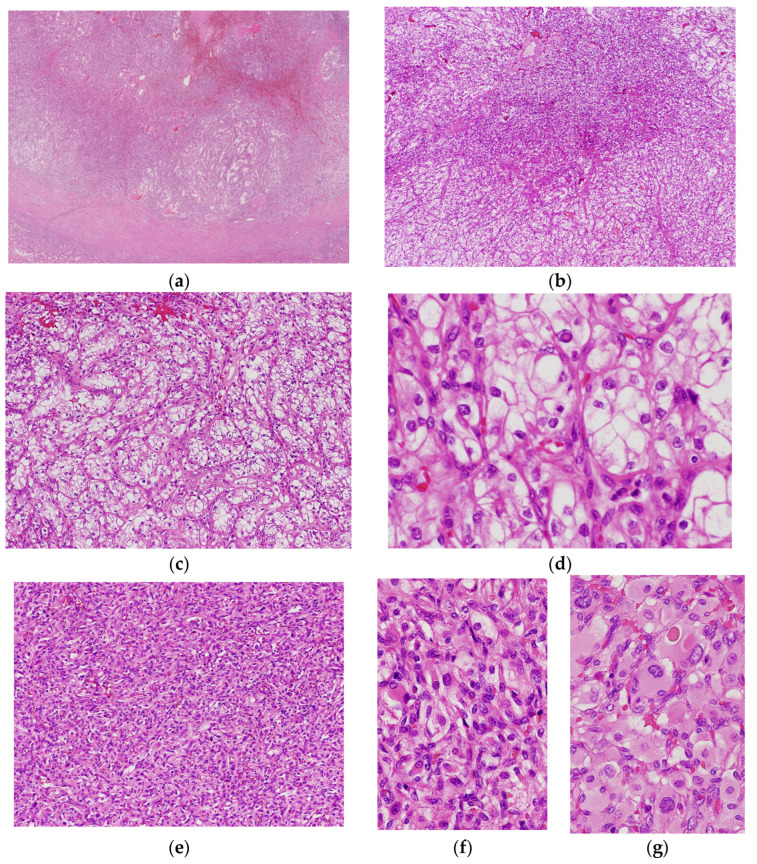
Histological findings of RCC with hemagioblastoma-like features. (**a**) Whole slide image displays the well-circumscribed tumor with a thick capsule. The tumor comprises CCRCC-like area in the periphery of the tumor and a HB-like area in the center. (**b**) Transitions between CCRCC- and HB-like areas are observed (low-power magnification). (**c**) In the low-power view of the CCRCC-like area, the tumor exhibited a solid nest and abortive tubular growth. (**d**) In the high-power view of the CCRCC-like area, tumor cells are composed of clear cells with abundant cytoplasm and mildly irregular nuclear contour without nucleolus. A few intranuclear inclusions are evident. (**e**) In the low-power view of the HB-like area, tumor cells exhibit sheet-like growth traversed with vessels. (**f**) Tumor cells are intimately intermixed with vessels, and the tumor cells are smaller and more indistinct than those in the CCRCC-like area (high-power magnification). (**g**) Tumor cells are large or small with abundant eosinophilic cytoplasm. Some present irregular nuclear contours. A hyaline globule was noted (high-power magnification).

**Figure 4 biomedicines-11-01467-f004:**
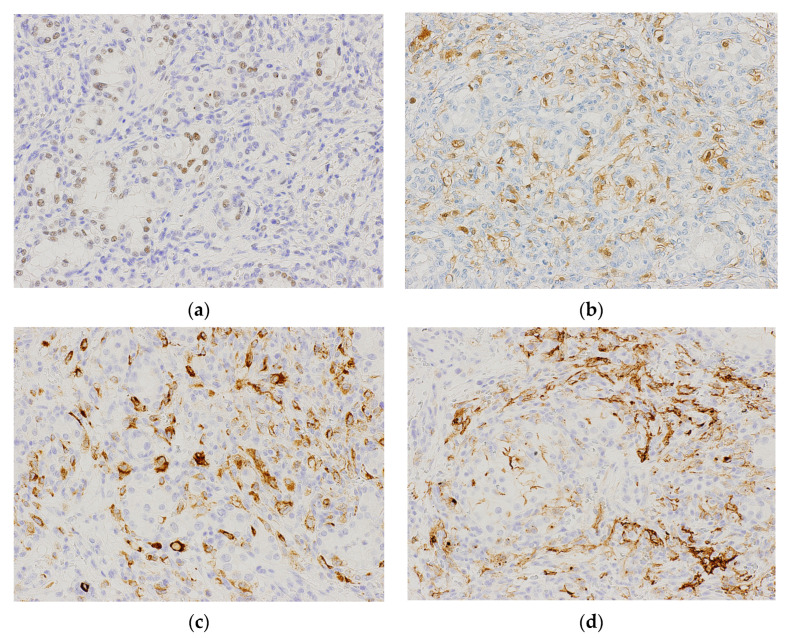
Immunohistochemical findings of RCC with hemangioblastoma-like features. (**a**) PAX8 is positive mainly at rigorous tumor nests (200× magnification). (**b**) Tumor cells exhibit CD10 positivity (200× magnification). Inhibin-α (**c**) and S100 (**d**) are positive in the HB-like area (200× magnification).

**Figure 5 biomedicines-11-01467-f005:**
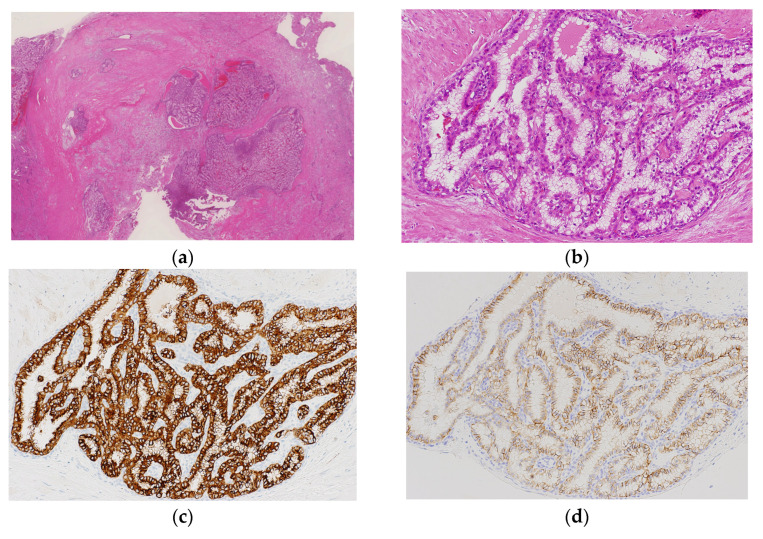
RCC with fibromyomatous stroma. (**a**) Whole slide image displays the well-circumscribed tumor with a thick capsule. A fibromuscular bundle extended from the capsule traversed the tumor and the tumor showed its multiple nodular appearance. (**b**) The tumor exhibits elongated tubular growth with a branch and small papillary tuft (100× magnification). (**c**) The tumor has diffuse, strong immunopositivity for CK7 (100× magnification). (**d**) The tumor has foci of cup-shaped immunopositivity for CAIX (100× magnification).

**Table 1 biomedicines-11-01467-t001:** Clinicopathological and molecular features of RHB, RCC-HB, and their related tumors.

	RHB (*n* = 29)	RCC-HB (*n* = 5)	RCC-FMS * [25,26,29]	CNS-HB [2]
age (in years) (range)	47 ^(1)^ (16–72)	38 ^(1)^ (32–75)	adult	adult
sex (man: woman)	1:1	2:3	1:1–2	1:1
sings of VHL	-	-	-	+
size (cm) (range)	3.5 ^(1)^ (1.2–8.0)	3.2 ^(1)^ (1.7–5.5)	1.0–4.5	<3.0
macroscopic findings	well-circumscribed solid	well-circumscribed solid	solid	cystic with mural solid nodule
histological findings	multivacuolated stromal cells with rich vascular network	CCRCC-like and HB-like area (with focal FMS)	branching tubules with focal papilla of clear cells and FMS	similar to that of RHB
immunohistochemical findings	Positive: inhibin-α (28/29), S100 (28/28), PAX8 (9/17), pan-CK (4/24), EMA (6/18), CD10 (8/19), CAIX (3/4) Negative: RCCm (0/4), CK7 (0/4)	Positive: PAX8 (4/5), CAIX (2/2), CD10 (4/5), EMA (2/2), CK (AE1/AE3) (2/2), CK (CAM5.2) (1/1), RCCm (2/3), CK7 (2/2) ^(2)^, inhibin-α (5/5) ^(3)^, S100 (4/4) ^(3)^ Negative: AMACR (0/1)	Positive: PAX8, CD10, CAIX, CK7, CK-HMW	Positive: inhibin-α, S100 Negative: pan-CK, RCCm, EMA, CD10
molecular	no *VHL*	TSC2/MTOR?, no *VHL*?	TSC/MTOR, no *VHL*	*VHL*
prognosis	indolent	indolent ^(4)^	indolent	indolent

RHB, renal hemangioblastoma; RCC-HB, renal cell carcinoma with hemangioblastoma-like features; RCC-RMS, renal cell carcinoma with fibromyomatous stroma; CNS-HB, hemangioblastoma of the central nervous system; VHL, Von Hippel–Lindau disease; CCRCC, clear cell renal cell carcinoma; HB, hemangioblastoma; FMS, fibromyomatous stroma; RCCm, RCC-marker; CK-HMW, CK high molecular weight. * RCC-FMS, including two *ELOC*-mutated renal cell carcinomas. ^(1)^ Median age or size. ^(2)^ CK7 was negative in the HB-like area. ^(3)^ Inhibin-α and S100 were negative in the CCRCC-like area. ^(4)^ One RCC-HB was pT3a and the patient had not been followed up.

## Data Availability

Data are available in a publicly accessible repository.

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
