# Peer review of "Current Knowledge and Prospects for Renal Hemangioblastoma and Renal Cell Carcinoma with Hemangioblastoma-like Features"

_biomedicines, 2023, doi:10.3390/biomedicines11051467_

Round 1

Reviewer 1 Report

This is a well conducted and straightforward review of Renal Hemangioblastoma and Renal Cell Carcinoma with Hemangioblastoma-like Features. The text is clear, the pictures of good quality. This paper should be of interest for most people in the field. No specific comment from my part. Please specify the meaning of TSC (line 181 tuberous sclerosis ? ).

The last sentence of the conclusion is somewhat elusive on the nature of the future studies than should help "to elucidate the relationships between RHB, RCC-HB, CNS-HB, and RCC-FMS". Based on their expertise, could the authors mention briefly what kind of combined approach might be the most appropriate to enhance the knowledge on this spectrum of tumors.

Author Response

Thank you very much for your review.

We revised our manuscript as follows according to your recommendation.

  1. Line 186 → 180
  2. Line 258–261 → 252–255. 

Reviewer 2 Report

Authors Kojima et al have presented the pathological findings and differentials along with the latest clinical IHC panels which can help distinguish hemangioblastoma from its morphological mimicker Clear cell renal cell carcinoma. In addition, they have paid some attention to an emerging entity called renal cell carcinoma with hemangioblastoma-like features. Although a good pathological read but it would have been nicer if a comprehensive molecular/genomic background features would have been touched upon apart from the newer advances and concepts leading up to therapeutics and outcomes. A more comprehensive tables or algorithms for lots of details provided is also highly recommended. Right now, the information in the "opinion" section is neither novel presented, nor it is a comprehensive yet concise amalgamation of newer concepts. I am willing to re-assess if the manuscript is spruced up with the suggestions and modified accordingly. 

Is acceptable minor changes needed. 

Author Response

Thank you very much for your review. 

We revised our manuscript as follows according to your recommendation.

1) We summarized the molecular features and the presumed molecular features of RHB, RCC-HB, RCC-FMS, and CNS-RHB.

2) We provided comprehensive table after "Conclusion".

Round 2

Reviewer 2 Report

Could have been a bit more comprehensive but accept authors opinions and efforts to touch upon the rare entities. Best wishes. 

Minor edits required.